# Thermodynamics and Kinetics of Glycolytic Reactions. Part I: Kinetic Modeling Based on Irreversible Thermodynamics and Validation by Calorimetry

**DOI:** 10.3390/ijms21218341

**Published:** 2020-11-06

**Authors:** Kristina Vogel, Thorsten Greinert, Monique Reichard, Christoph Held, Hauke Harms, Thomas Maskow

**Affiliations:** 1UFZ–Helmholtz Centre for Environmental Research, Department of Environmental Microbiology, Leipzig, Permoserstr. 15, D-04318 Leipzig, Germany; Kristina.vogel@ufz.de (K.V.); monique.reichard@ufz.de (M.R.); Hauke.harms@ufz.de (H.H.); 2Institute for Drug Development, Leipzig University Medical School, Leipzig University, Bruederstr. 34, 04103 Leipzig, Germany; 3Laboratory of Thermodynamics, Department of Biochemical and Chemical Engineering, Technische Universitaet Dortmund, Emil-Figge-Str. 70, 44227 Dortmund, Germany; thorsten.greinert@tu-dortmund.de (T.G.); christoph.held@tu-dortmund.de (C.H.)

**Keywords:** biothermodynamics, glycolysis, isothermal titration calorimetry, systems biology

## Abstract

In systems biology, material balances, kinetic models, and thermodynamic boundary conditions are increasingly used for metabolic network analysis. It is remarkable that the reversibility of enzyme-catalyzed reactions and the influence of cytosolic conditions are often neglected in kinetic models. In fact, enzyme-catalyzed reactions in numerous metabolic pathways such as in glycolysis are often reversible, i.e., they only proceed until an equilibrium state is reached and not until the substrate is completely consumed. Here, we propose the use of irreversible thermodynamics to describe the kinetic approximation to the equilibrium state in a consistent way with very few adjustable parameters. Using a flux-force approach allowed describing the influence of cytosolic conditions on the kinetics by only one single parameter. The approach was applied to reaction steps 2 and 9 of glycolysis (i.e., the phosphoglucose isomerase reaction from glucose 6-phosphate to fructose 6-phosphate and the enolase-catalyzed reaction from 2-phosphoglycerate to phosphoenolpyruvate and water). The temperature dependence of the kinetic parameter fulfills the Arrhenius relation and the derived activation energies are plausible. All the data obtained in this work were measured efficiently and accurately by means of isothermal titration calorimetry (ITC). The combination of calorimetric monitoring with simple flux-force relations has the potential for adequate consideration of cytosolic conditions in a simple manner.

## 1. Introduction

In systems biology, models are expected to describe enzyme kinetics with high precision at relatively low metabolite concentrations (in the lower micro- to millimolar range). Since all metabolic fluxes decay when reaching a thermodynamic equilibrium, the approximation to the state of equilibrium must also be well described. Often, only the kinetic data of enzyme-catalyzed reactions are published. Based on such data, various kinetic models for data analysis of enzyme-catalyzed reactions have been developed in the literature. Figure 1 compares several of such kinetic models with regard to the validity range (A) and the required adjustable parameters (B). The most common approach is Michaelis-Menten kinetics, which is based on the publications of Brown [1] and Henri [2,3]. Leonor Michaelis and Maud Menten studied the kinetics of the invertase reaction in 1913 [4]. They defined the initial velocity of a reaction *r* (in mol L^−1^ s^−1^) (Equation (1)) as a function of the concentrations of the substrate $c_{S}$*,* the product $c_{P}$*,* and the enzyme $c_{E}$ (all in mol L^−1^). Note that the product formation rate corresponds to the substrate consumption rate with an inverted sign:(1)$$r=\frac{{d\;c}_{P}}{\mathrm{dt}}=-\frac{{d\;c}_{S}}{\mathrm{dt}}=\frac{r_{\max}\cdot c_{S}}{K_{S}{+c}_{S}}$$
with:(2)$$r_{\max}{=k}_{\mathrm{cat}}\cdot c_{E}$$

$k_{cat}$ (in s^−1^) is a kinetic constant of first order. $r_{max}$ stands for the maximum reaction rate (in mol L^−1^ s^−1^). $K_{S}$ is called Michaelis constant and is the substrate concentration at which the reaction rate is half of $r_{max}$. The Michaelis-Menten equation is based on the following reaction mechanism (Equation (3)):(3) E+ S  ←k−1→k1ES→kcatE+P 

Here, an enzyme (*E*) binds reversibly to a substrate (*S*) and forms an enzyme-substrate-complex (*ES*) that releases the enzyme and the product (*P*) in an irreversible step. $k_{1}$ and $k_{-1}$ are the reaction rate constants for the association and the dissociation of the enzyme-substrate-complex, respectively and $k_{cat}$ describes the dissociation of the enzyme-substrate-complex to the product and the enzyme. However, this model representation for enzyme kinetics is only valid for irreversible reactions. In reversible enzymatic reactions, the substrate is only consumed until the equilibrium is reached and the reaction mechanism has to be extended (Equation (4)) for the backward reaction, which is characterized by the kinetic parameter $k_{-cat}$ [5]:(4) E+ S  ←k−1→k1ES  ←k−cat→k cat E+P 

For simplicity, the majority of published research focused on the initial steady state of reactions (no product formation), so that the backward reaction could be neglected [6]. This simplification allows applying the irreversible Michaelis-Menten model also to reversible reactions. However, many enzymatic reactions are highly reversible [7,8,9,10,11], and the complete reaction progress cannot be represented if the backward reaction is neglected [12,13]. There are some models which incorporate the reaction equilibrium into Equation (1), such as the equilibrium-based model of Hoh and Cord-Ruwisch [14] and the reversible Michaelis-Menten mechanism [6,11,12,13,15,16,17]. The reversible Michaelis-Menten mechanism relies on a complicated system of kinetic equations. It requires a high number of kinetic constants, which are not independent of each other (e.g., for an isomerization mechanism, four parameters are obtained and five independent measurements are needed to determine them) [14,16,18]. Furthermore, the high number of parameters makes it difficult to predict or even describe the influence of the cytosolic conditions on reaction kinetics as changes to the individual parameters could cancel each other out. Therefore, alternative ways for data analysis of reversible reactions with a focus on determining the influence of cytosolic conditions on reaction kinetics are required. Simple linear flux-force relations derived from irreversible thermodynamics need only one parameter (called the phenomenological coefficient) to describe the approximation to the state of equilibrium. Note that linear relations are only to be expected in relative proximity to the equilibrium. In order to validate this approach, we have chosen two reactions from glycolysis as an example because glycolysis is among the best-understood pathways. Thus, in the present work, we will show the usefulness of the flux-force relations from irreversible thermodynamics to describe the reaction behavior of both glycolytic reactions, and we will validate the results by a series of new reaction data. The chosen glycolytic reactions under investigation in this work are catalyzed by glucose-6-phosphate isomerase (reaction 2 of glycolysis, Equation (5)) and by phosphopyruvate hydratase (enolase) (reaction 9, Equation (6)). Both reactions are reversible and end in thermodynamic equilibrium. The first reaction is the conversion of glucose-6-phosphate (G6P) into fructose-6-phosphate (F6P) and the second reaction is the transformation of 2-phosphoglycerate (2PG) into phosphoenolpyruvate (PEP) and water:(5)G6P⇆F6P 
(6)$$2\mathrm{PG}⇆\mathrm{PEP}+H_{2}O$$

Systematic analyses of the published thermodynamic and kinetic studies of both reactions reveal two shortcomings. First, the kinetic data are mainly evaluated using the irreversible Michaelis-Menten model and therefore, disregard the reversibility of the reaction [19,20,21,22,23,24,25]. Second, the measurement conditions chosen by the researchers often considerably deviate from the environment inside biological cells. However, numerous studies show that environmental conditions (e.g., the solvent) have a considerable influence on both the equilibrium and the kinetics of reversible enzyme reactions [26,27,28,29,30]. Thus, cell-mimicking conditions should be taken into account for a realistic thermodynamic and kinetic investigation of metabolic pathways [31,32]. One aim of this work was to examine to what extent models derived from irreversible thermodynamics allow a satisfactory description of the full reaction progress and especially the approximation to the equilibrium state. This is important because in metabolic reaction sequences, such as glycolysis, many reactions occur at metabolome concentrations in the milli- and micromolar range and close to the equilibrium (Figure 1A). Two models will be applied in the present work. The first model was suggested by Noor 2013 and is called ‘separable rate law’ [9]. It breaks down the reversible Michaelis-Menten rate law into a product of three factors: (i) The maximum possible conversion rate, (ii) the enzyme saturation level, and (iii) the thermodynamic driving force. The second model (a flux-force model) was published by Beard and relates fluxes with the thermodynamic driving force [9,33]. The temperature dependence of the rate constants until deactivation of the enzyme is best described with models (e.g., Arrhenius equation). Therefore, the temperature dependence in addition to the exact description of the reaction process was used to check the plausibility of the tested kinetic models (Figure 1C).

Any of the above-described model still needs experimental data. Thus, new data are presented in this work. Calorimetry provides real time information about the investigated reaction, requiring neither sampling nor interacting with the reaction partners [34]. Therefore, isothermal titration calorimetry (ITC) in combination with thermokinetic data evaluation, developed by Todd and Gomez [35], was used to monitor the reaction progress and its dependency on the temperature. Our final goal is to develop a model that allows describing the influence of cytosolic conditions on the kinetics of enzymatic reactions.

## 2. Results

### 2.1. Definitions and Specifications

In the following, all concentrations are given as molality (mol kg^−1^) instead of molarity (mol L^−1^). The latter is more common in biochemistry and bioengineering but the use of molality is important for the link to thermodynamics. Molality only depends on the masses of solute and solvent, which are unaffected by variations in temperature, pressure, and density. From a practical point of view, the differences between molality and molarity under cytosolic conditions are small. For instance, the density of water decreases by only 0.6% in the temperature range between 25 and 40 °C. Table 1 shows a list of all symbols used, their properties and units.

### 2.2. Application of Irreversible Thermodynamics for Kinetic Evaluation

Irreversible thermodynamics postulates a linear relation between flux or reaction rate *r* (mol kg^−1^ s^−1^) and the “conjugate” thermodynamic driving force ι if the system is not too far from the equilibrium (Equation (7)). ι is zero at equilibrium, positive for the forward reaction, and negative for the backward reaction [36]:(7)r = ∝·ι
with
(8)$${\iota \;=\;1-e}^{\Delta_{R}G/\mathrm{RT}}$$
and
(9)$$\Delta_{R}G=\Delta_{R}G^{o}+R\cdot T\cdot \ln\left(\underset{i}{\prod }a_{i}^{\nu_{i}}\right)$$

*α*, the correlation factor, links the substrate consumption rate *r* with the thermodynamic driving force. $\Delta_{R}G$, $\Delta_{R}G^{o}$, $a_{i}$, and $\nu_{i}$ are the Gibbs energy and the standard Gibbs energy of the reaction (both in J mol^−1^), the activity and the stoichiometric coefficients, respectively. *R* and *T* are the universal gas constant and the temperature. In both investigated reactions we evaluated the opposite direction of the glycolysis; that is, F6P is considered as a substrate and G6P as a product of reaction 2, while PEP is considered as a substrate and 2PG as a product of reaction 9. Taking the equilibrium condition ($\Delta_{R}G^{o}\;=\;-R\cdot T\cdot ln\left(K_{a}\right)$) and assuming that the activities of the solvent during the reaction and at equilibrium are identical, leads to the following result for the Gibbs energy (Equation (10)) and for the thermodynamic driving force (Equation (11)):(10)$$\Delta_{R}G^{\prime }\;=\;R\cdot T\cdot \ln\left(\frac{a_{S}^{\mathrm{eq}}\cdot a_{P}}{a_{S}\cdot a_{P}^{\mathrm{eq}}}\right)$$
(11)$$\iota =\frac{a_{S}\cdot a_{P}^{\mathrm{eq}}{-a}_{S}^{\mathrm{eq}}\cdot a_{P}}{a_{S}\cdot a_{P}^{\mathrm{eq}}}$$
where $a_{S}^{eq}$, $a_{P}$,$\;a_{S}$, and $a_{P}^{eq}$ stand for the equilibrium activity of the substrate, the activity of the product, the activity of the substrate, and the equilibrium activity of the product, respectively.

This model was refined by Noor in 2013 [9] who expressed α by the product of the maximum rate called the capacity term *V^+^* and the enzyme fractional saturation term level *κ* (Equation (12)):(12)$$r\;=\;\propto \cdot {\iota \;=\;V}^{+}\cdot \kappa \cdot \iota$$
(13)$$V^{+}{=c}_{E}\cdot k_{\mathrm{cat}}^{+}$$


(14)$$\kappa =\frac{c_{S}/K_{S}}{1+c_{S}/K_{S}+c_{P}/K_{P}}$$


$k_{cat}^{+}$ (in s^−1^), $c_{S}$,$\;c_{P}$, $K_{S}$, and $K_{P}$ (in mol kg^−1^) denote the kinetic constant of the forward reaction, the substrate concentration at any time of the reaction, the product concentration at any time of the reaction, and the Michaelis constants for the substrate and product, respectively. Using Equations (10) and (12)–(14), the mass balance ($c_{P}\;=\;c_{S}^{0}-c_{S}$) and neglecting the activity coefficient Equation (15) can be formulated for the description of the enzymatic reactions.
(15)$${r\;=\;c}_{E}\cdot k_{\mathrm{cat}}^{+}\cdot \frac{\frac{c_{S}}{K_{S}}}{1+\frac{c_{S}}{K_{S}}+\frac{\left(c_{S}^{0}{-c}_{S}\right)}{K_{P}}}\cdot (1-\frac{c_{S}^{\mathrm{eq}}\cdot \left(c_{S}^{0}{-c}_{S}\right)}{\left(c_{S}^{0}{-c}_{S}^{\mathrm{eq}}\right)\cdot c_{S}})$$

Here, $c_{S}^{0}$ is the concentration of the substrate at time 0. In the original concept of Noor, an enzyme saturation by (i) the substrate, (ii) the product, or (iii) by both is allowed. We found the Noor equation suitable to describe our data with only two adjustable parameters assuming a single saturation of either the substrate or the product. For reaction 2, we investigated an enzyme saturation by the substrate and the adjustable parameters are $r_{max}$ and $K_{F6P}$. For reaction 9, an enzyme saturation by the product was investigated and the adjustable parameters are $\Lambda \;{=\;c}_{E}\;\cdot \frac{k_{\mathrm{cat}}^{+}}{K_{\mathrm{PEP}}}$ and $K_{2PG}$ by excluding the double saturation (Equations (16) and (17)). The determination of $r_{max}$ for reaction 9 was not possible because $k_{cat}^{+}$ is not independent from $K_{PEP}$. The parameters were obtained by a non-linear regression of *r* vs. $c_{S}$.
(16)$$\mathrm{For}\;\mathrm{reaction}\;2:\;r\;=\;r_{max}\cdot \frac{\frac{c_{F6P}}{K_{F6P}}}{1+\frac{c_{F6P}}{K_{F6P}}}\cdot (1-\frac{c_{F6P}^{\mathrm{eq}}\cdot \left(c_{F6P}^{0}{-c}_{F6P}\right)}{\left(c_{F6P}^{0}{-c}_{F6P}^{\mathrm{eq}}\right)\cdot c_{F6P}})$$
(17)$$\mathrm{For}\;\mathrm{reaction}\;9:\;r\;=\;\Lambda \cdot \frac{c_{\mathrm{PEP}}}{1+\frac{\left(c_{\mathrm{PEP}}^{0}{-c}_{\mathrm{PEP}}\right)}{K_{2\mathrm{PG}}}}\cdot (1-\frac{c_{\mathrm{PEP}}^{\mathrm{eq}}\cdot \left(c_{\mathrm{PEP}}^{0}{-c}_{\mathrm{PEP}}\right)}{\left(c_{\mathrm{PEP}}^{0}{-c}_{\mathrm{PEP}}^{\mathrm{eq}}\right)\cdot c_{\mathrm{PEP}}})$$

The second model we tested was introduced by Westerhoff and Beard and is called the flux-force relationship [9,33]. They combined ${∆}_{R}G$ with fluxes for the forward *J^+^* and the backward reaction *J^−^* and Noor substituted the fluxes by the respective rates [9] (Equation (18)). A neatly deduction of Equation (18), can be found in the publication by Beard and Qian [33]:(18)$${∆}_{R}G\;=\;-R\cdot T\cdot {\ln(J}^{+}/J^{-})\;=\;-R\cdot T\cdot \ln\left(\frac{r^{+}}{r^{-}}\right)$$

The net reaction rate *r* is the difference between reaction rates of the forward $r^{+}$ and backward reaction $r^{-}$. This can be related to the Gibbs energy (Equation (19)) and the thermodynamic driving force ı:(19)$$\frac{r}{r^{+}}=\frac{r^{+}-r^{-}}{r^{+}}=1-\exp\left({∆}_{R}G/R\cdot T\right)\;=\;\iota$$

Both, forward and backward reaction use the same catalytic center of the enzyme and it is justified to assume a proportionality between the sum of *r*^+^ and *r*^−^ and the amount of enzyme $c_{E}$ [9,37]. Thus, the ratio between the net rate and the sum of *r*^+^ and *r*^−^ needs to be evaluated (Equation (20)).
(20)$$\frac{r}{r^{+}{+r}^{-}}\;=\;\frac{r^{+}{-r}^{-}}{r^{+}{+r}^{-}}\;=\;\frac{1-\frac{r^{-}}{r^{+}}}{1+\frac{r^{-}}{r^{+}}}\;=\;\frac{{1-e}^{{∆}_{R}G/\mathrm{RT}}}{{1+e}^{{∆}_{R}G/\mathrm{RT}}}\;=\;\frac{\iota }{2-\iota }$$

Now, we introduce that the sum of *r^+^* and *r^−^* is proportional to $c_{E}$ into Equation (20), an expression for the rate dependency on the thermodynamic driving force results (Equation (21)). The proportionality factor *L* is called a phenomenological coefficient or kinetic parameter in the following:(21)$$r\;=\;\left(r^{+}{+r}^{-}\right)\cdot \frac{\iota }{2-\iota }\;=\;L\cdot c_{E}\cdot \frac{\iota }{2-\iota }$$

The great biggest advantage of this model is that it correlates the rate with the substrate concentration as the frequently used Michaelis-Menten model does, but contains only one fit parameter *L*, which is a kinetic parameter that can be very suitable for further investigations aiming at comparing the influence of cytosolic conditions on the kinetics. Furthermore, it perfectly describes the approximation to the equilibrium state. For data analysis, a plot of the rate against $\frac{\iota }{2-\iota }$ is used and the slope of a linear regression of the measured data provides (at known enzyme concentration) the value for *L*. Since the measured heat flow data at the beginning of the reactions are disturbed by the inertia of the ITC and the heat of dilution, this part of the curve will not be used for kinetic evaluation. This corresponds to $\frac{\iota }{2-\iota \;}>\;0.6$ for both reactions.

Typically, the Arrhenius equation (Equation (22)) describes the temperature dependence of the kinetic constants below the optimal temperature very well. Therefore, it was verified whether the parameter *L* of the flux-force model follows the Arrhenius relationship [38]:(22)$$L\;=\;A\;\cdot {\;e}^{-\frac{E_{a}}{R\cdot T}}$$

*A* is the pre-exponential factor (a constant value). The activation energy $E_{a}$ can be calculated from the slope of a plot of ln k vs. 1/T. Experimental data are needed to evaluate the kinetic models and the temperature dependency of the kinetic constant.

### 2.3. ITC Results

In this work, ITC was used to monitor the reaction progress as a basis to determine the kinetic parameters. The parameters $K_{S},\;K_{P}$, $r_{max}$, and Λ of the Noor model and the kinetic parameter *L* were determined at different temperatures to access their influence on the kinetics of the reaction. The reaction progress was monitored calorimetrically and the enthalpy of reaction as well as the kinetic constants were determined. The measuring principle for reactions 2 and 9 is illustrated in Figure 2A,C. In Figure 2A,C, the raw data from the reaction signal (green), reference signals from substrate to buffer (Figure 2A,C, red) and from buffer to enzyme (Figure 2A,C, black) are shown. The second reference signal was negligibly small. The net signal is the difference between the raw data and the reference measurement. To determine the heat *Q* the signal was integrated (Figure 2B,D). The heat obtained from reactions 2 and 9 is marked in gray in Figure 2B,D.

In order to exclude that the ionization of the buffer plays a role for the obtained reaction enthalpy, the measurements were also performed in other buffers [39]. For reaction 2, a potassium phosphate buffer and a HEPES buffer were used for this purpose. The differences in the ionic strengths of the buffers were neglected. The reaction enthalpy values are all very similar (between 9.6 and 11.1 kJ mol^−1^) (Table 2), so we concluded that the ionization of the buffer does not play a significant role for the measured values. The small deviations of the values could be caused by the influence of the ionic strength. The same results were previously obtained for reaction 9 [40].

To determine the reaction enthalpy, the heat *Q* from the ITC and equilibrium concentrations of the substrates are required (Equation (23)). With the help of the Perturbed-Chain Statistical Associating Fluid Theory (ePC-SAFT), equilibrium concentration ratios $K_{c}$ can be calculated from the already known thermodynamic equilibrium constants $K_{a}$. The parameters are listed in Table 3. It can be seen that a temperature change in the investigated area has no influence on the calorimetrically determined heat. The equilibrium $K_{c}$ values of both reactions increase with the increasing temperature. For reaction 2, from 0.285 (298.15 K) to 0.343 (310.15 K). For reaction 9, $K_{c}$ increases from 239.4 (298.15 K) to 251.3 (310.15 K).

### 2.4. Kinetic Analyses

The net heat flow values for both investigated reactions have been transformed into reaction rates and substrate concentrations using the Todd and Gomez concept Equations (29) and (30) [35]. Surprisingly, the reaction rate does not show the typical saturation behavior of Michaelis-Menten kinetics (Figure 3). Furthermore, the reaction rate approaches zero when the equilibrium concentration of the substrates are reached. Additionally, the curve of reaction 9 (Figure 3B) shows a convex curvature, which Michaelis-Menten cannot describe. The convex behavior comes from the enzyme saturation by the product.

The first model that we have tested was published in 2013 by Noor et al. [9]. The Noor-model seems to be much better suited than the Michaelis-Menten because it is equilibrium based. For both, reactions 2 and 9, the Noor model (red) fits very well to our measured data (black dots) (Figure 4). The initial phase of the measurements, which was affected by the heat of dilution and inertia of the ITC, was not used for data analysis. As shown in Figure 2, the curves of both reactions have a different curvature. The Noor model can fit both of these curvatures. Reaction 2 has a concave curvature (Figure 4A), for which the Noor assumption of an enzyme saturation by the substrate F6P applies [9]. In this case, Equation (16) was used for the fit. Reaction 9 has a convex curvature (Figure 4B), for which a different Noor assumption of enzyme saturation by the product 2PG [9] was used. Equation (17) was applied in this case. This results in two fit parameters for both reactions: For reaction 2, we obtain $r_{max}$ and $K_{F6P}$ and for reaction 9, we obtain Λ and $K_{2PG}$.

The results of the Noor analysis are presented in Table 4. In reaction 2 of glycolysis, an increase of $r_{max}$ can be seen with the increasing temperature, whereas the $K_{F6P}$ value shows a decrease. In contrast, in reaction 9 both, Λ and $K_{2PG}$, increase with the increasing temperature.

Additionally, the one-parameter flux-force relationship model of Beard [33] was applied in this work. In Figure 5, the linearized flux-force model is presented, and the slope corresponds to the product of the enzyme concentration and the kinetic parameter *L* (Equation (21)).

Figure 5 demonstrates that the flux-force model fits the data well. With flux-force the range that cannot be used for data evaluation ($\frac{\iota }{2-\iota }>0.6$) is larger than with Noor. The reason for this is that the flux-force model can best reproduce the data near the equilibrium. The data generated for both reactions can be found in Table 5. In both reactions a strong increase of the *L* value can be seen with the increasing temperature.

Both of the investigated models are able to reproduce the calorimetrically measured data well. The next question is, whether the kinetic parameters from the flux-force model fulfill the Arrhenius equation. For the Arrhenius equation, the natural logarithm of the respective rate constant is plotted against the reciprocal temperature in Kelvin [38]. The activation energy can then be determined from the slope of the linear fit (Equation (22)). Figure 6 shows that the obtained kinetic parameter *L* can be described by the Arrhenius model for both reactions. Under the measuring condition, an activation energy of 55.4 ± 1.2 kJ mol^−1^ was calculated for reactions 2 and 9 the value was 44.8 ± 1.4 kJ mol^−1^.

## 3. Discussion

Our aim was to evaluate whether different kinetic models are able to reproduce the kinetics of reversible reactions close to the equilibrium. Therefore, we tested two models, Noor and flux-force, and determined whether these models can reflect the temperature dependence using an Arrhenius equation. We analyzed and tested the models by comparing them to new kinetic reaction data for the glycolytic reactions 2 and 9 (PGI and enolase), respectively.

### 3.1. Thermodynamic Data for Reaction 2

The data obtained can be found in Table 3 and Figure 2. At 310.15 K, a reaction enthalpy of ${∆}_{R}H$ = 11.1 ± 0.5 kJ mol^−1^ was obtained for the considered conditions. Comparable values can be found in the literature. In 1968, standard reaction enthalpies of 10.8 kJ mol^−1^ (273.15–308.15 K) and 15.2 kJ mol^−1^ (308.15–322.15 K) were received from Dyson et al. [24]. Three explanations for the deviations from our data are possible: (i) ${∆}_{R}H$ might be strongly temperature dependent (the reaction heat capacity is not zero), (ii) an insufficient consideration of the physical environment (activity coefficients), which is the reason that ${∆}_{R}H$ might be strongly dependent on the reaction conditions (substrate concentrations, solvent, and cosolvents), and (iii) the different origins of the enzyme (Dyson: Rabbit skeletal muscle), which might also contain impurities that further influence ${∆}_{R}H$. In 1988 Tewari published a calorimetric standard reaction enthalpy ${∆}_{R}H$^0^ = of 11.7 ± 0.2 kJ mol^−1^ at 298.15 K [41]. A more recent calorimetric value of ${∆}_{R}H$^0^ = 12.05 ± 0.2 kJ mol^−1^ and a van ‘t Hoff value for ${∆}_{R}H$^0^ = 12.25 ± 0.3 kJ mol^−1^ were published in 2014 by Hoffmann et al. [42]. Most literature values are standard reaction enthalpies. As explained in [40], these ${∆}_{R}H$^0^ values cannot be equated with the buffer-dependent values measured in the present work, as ${∆}_{R}H$^0^ denotes the standard state at infinite dilution and zero ionic strength. Therefore, a reaction enthalpy ${∆}_{R}H$ was calculated using van ‘t Hoff and the $K_{c}$ values were calculated using ePC-SAFT at exactly the conditions for the calorimetric experiments in this work. This amounts to ${∆}_{R}H$ = 11.9 kJ/mol, which perfectly agrees with the calorimetrically measured ${∆}_{R}H$^0^ value. This is an excellent result considering that the ePC-SAFT parameters for 2PG were inherited from 3PG (due to the inaccessibility to thermodynamic data for 2PG). The reaction enthalpy does not show any temperature dependence. This result does not fit to the measurements of Tewari who found a slight decrease of the reaction enthalpy with the increasing temperature [41]: ${∆}_{R}H$ decreased from −9.025 ± 0.0331 kJ mol^−1^ at 304.95 K to −9.144 ± 0.0052 kJ mol^−1^ at 316.15 K. The experimental uncertainty of the values from Tewari are very small, therefore, the experimentally observed small temperature dependency of ${∆}_{R}H$ from Tewari is hidden by the uncertainty of our calorimetric experiments.

### 3.2. Validation of Kinetic Models by the New Calorimetric Data

The heat production rates of reactions 2 and 9 of glycolysis were calorimetrically monitored to determine the kinetic parameters. In the literature, various kinetic models have been tested to describe the experimental data. The most common approach is the irreversible Michaelis-Menten model. For this purpose, initial reaction rates are investigated at which product formation is so low that it can be neglected. Indeed, reaction 2 [24,25,43,44,45] and reaction 9 [20,21,22,46,47,48] have already been kinetically investigated in previous works using Michaelis-Menten kinetics. In our investigations we are interested in the kinetics up to near the equilibrium. For that purpose, the irreversible Michaelis-Menten model is not useful. Further, the number of the required kinetic parameters using the reversible Michaelis-Menten kinetics is too high, i.e., it either requires a lot of experimental data or is underdetermined with the number of data available. Furthermore, our final goal is to develop a model that allows describing the influence of cytosolic conditions on the kinetics of enzymatic reactions. However, if several kinetic parameters are obtained for each condition under investigation, there may be problems in evaluating them, so that the influence of cytosolic conditions cannot be clearly identified from the data. Noor combined a kinetic approach with irreversible thermodynamics. The approach simplifies the total equation and the number of required parameters by discussing four different special cases which are characterized by different saturation effects [9]. We adapted all these cases to our data, and we found the following: For reaction 2, the case of saturation by the substrate and for reaction 9, the saturation by the product are most suitable to describe our data (Figure 4). This provides two fit parameters for both reactions: $r_{max}$ and $K_{F6P}$ for reaction 2, and Λ and $K_{2PG}$ for reaction 9. Thus, the Noor model reflects well our measured data (data are given in Table 4). As two parameters are required in that model, problems might arise with the transferability of these parameters to reaction conditions that cause different behaviors than those used in the fit ($r_{max}$ increases with the increasing temperature but the $K_{F6P}$ value decreases). Models with only a single parameter allow clearly allocating the influence of temperature or other conditions. Further, the Noor model has another disadvantage: The obtained parameter Λ is not a kinetic constant in the conventional sense ($\Lambda \;=\;c_{E}\;\cdot \frac{k_{cat}^{+}}{K_{PEP}}$) since it contains $K_{PEP}$. In the Arrhenius plot (ln Λ vs. 1/T) we indeed found a linear temperature dependence (data not shown), but this argument might be too weak in favor of the Noor model. Thus, we analyzed whether the flux-force relationship is able to describe the experiments. In 2007, Beard suggested a relation between the rate of a reaction to its thermodynamic driving force, called the flux-force relationship (Equation (21)) [9,33]. This model describes our data with similar accuracy compared to the Noor model (Figure 5). A slight deviation is observed for high rates (directly after the start of the reaction) where the signal is potentially influenced by the thermal inertia of the calorimeter. Unfortunately, the flux-force model allows describing a smaller range of conditions than the Noor model (see Figure 1). The reason behind this is that the flux-force model can best reproduce the data near the equilibrium, and the beginning of the measurement was far away from the equilibrium. Therefore, the data at the beginning are neglected in the evaluation using the flux-force relationship. In sum, the big advantage of the flux-force relation is that only a single parameter is required to describe the kinetics. Such a single parameter can be compared very easily and the influence of the individual cytosolic conditions can be attributed to the changes in the parameter *L* directly.

### 3.3. Temperature Dependency of the Kinetics of Reaction 2

The temperature dependency of the kinetic parameter *L* can be described using the Arrhenius model (Figure 6). According to Equation (22), the activation energy can be calculated from the slope of the linear fit. Figure 6A shows that the data of reaction 2 fulfill the Arrhenius equation and an activation energy of $E_{a}\;$= 55.4 ± 1.2 kJ mol^−1^ was obtained. Following the van ‘t Hoff rule that a temperature rise of 10 K doubles the reaction rate [49], an activation energy of $E_{a\;}$ = 52.9 kJ mol^−1^ is estimated for the temperature rise from 298.15 to 308.15 K supporting our finding. In the investigated temperature range (between 298.15 and 310.15 K) we did not observe any deactivation, while measurements at 315.15 K showed a deactivation (data not shown). An analysis can only be performed in a temperature range where the enzyme is not deactivated. A comparable result was published by Dyson et al., which has determined a maximum reaction rate at 313.15 K followed by a reduction in the kinetic constant [24]. Two different values for the activation energy can be found in the paper from Dyson: $E_{a}\;$= 42.0 kJ mol^−1^ (288.15–303.15 K) and $E_{a}\;$= 22.5 kJ mol^−1^ (303.15–320.15 K) [24]. The values of Dyson were determined using $r_{max}$ with the already above-discussed problems. Since our parameter *L* is not the same as $r_{max}$, we get a slightly different value but still in the same order of magnitude for the activation energy. In addition, the investigations from Dyson were carried out with PGI from the rabbit skeletal muscle, which can also lead to different values for the activation energy. Another value was published by Sangwan with 31.2 kJ mol^−1^ (from $r_{max}$) for the enzyme in amyloplasts of immature wheat endosperm [50].

### 3.4. Temperature Dependency of Reaction 9

For reaction 9, an activation energy of $E_{a}\;$= 44.8 ± 1.4 kJ mol^−1^ was obtained. This value is also in the same range as the value of the van‘t Hoff rule. In 1957, Westhead published slightly higher activation energies for yeast enolase in a phosphate buffer (pH 6.8) of $E_{a}\;$= 61.1 kJ mol^−1^ and in a Tris/HCl buffer (pH 7.8) of $E_{a}\;$= 59.7 kJ mol^−1^ [22]. The deviations from our value can be explained by the different measuring conditions (e.g., pH and buffer). With both models (Noor and flux-force), we observed an increase of Λ and *L* with the increasing temperature. Our observed temperature dependency is consistent with a further study from Westhead, who only investigated a temperature range up to 305 K [51]. From this data, the activation energy can be calculated to be $E_{a}\;$= 55.5 kJ mol^−1^. Another work also reports a similar temperature behavior at elevated temperatures for an octameric thermophilic enolase [52].

## 4. Materials and Methods

### 4.1. Chemicals

Yeast enolase, monosodium phosphoenolpyruvate, and phosphoglucose isomerase type III from the baker’s yeast were purchased from Sigma-Aldrich (Sigma-Aldrich Chemie GmbH, Steinheim, Germany), fructose 6-phosphate disodium salt were from Alfa Aesar (Thermo Fisher (Kandel) GmbH, Kandel, Germany), MOPS was from AppliChem (AppliChem GmbH, Darmstadt, Germany), sodium chloride was from CHEMSOLUTE (Th. Geyer GmbH & Co. KG, Renningen, Germany), and magnesium chloride hexahydrate and sodium hydroxide were from Roth (Bernd Kraft, Duisburg, Germany). An overview about all the used chemicals, the CAS-numbers, and purity is given in Appendix A.

### 4.2. Sample Preparation for ITC Measurement

For every investigated reaction, two solutions, titrand and titrant, had to be prepared for the ITC measurements. The reaction buffer used for both investigated reactions consists of a 0.2 mol kg^−1^ MOPS buffer with 0.15 mol kg^−1^ Na^+^-ions (from sodium hydroxide and sodium chloride), 0.001 mol kg^−1^ MgCl_2_ at pH 7. The measurements were carried out at different temperatures to test whether the results obtained were in accordance with the Arrhenius equation. For reaction 2 of the glycolysis, the enzyme phosphoglucose isomerase (PGI) was obtained as a solution with 3 M ammonium sulfate. In order to avoid possible errors caused by the high ammonium sulfate concentration, the ammonium sulfate was removed. For this purpose, the sample was centrifuged at 10,000× *g* for 10 min at 2 °C. The obtained pellet contains the enzyme, while the supernatant containing the ammonium sulfate was removed. The pellet was dissolved in a reaction buffer that was adjusted to an enzyme concentration of 15 nmol kg^−1^. The substrate solution was prepared by dissolving F6P in the buffer to a concentration of 100 mmol kg^−1^. For reaction 9 of the glycolysis, the solutions were prepared as described in [40]. The enzyme solution had a concentration of 12 µmol kg^−1^ enolase and the substrate solution had a concentration of 89.5 mmol kg^−1^ PEP. Since PEP is a weak acid, the used MOPS-buffer could not keep the pH at 7. Therefore, an adjustment of the pH of the PEP with a 10 mol kg^−1^ sodium hydroxide solution was necessary. To not alter the buffer concentrations of the PEP solution, sodium hydroxide was also dissolved in the reaction buffer.

### 4.3. ITC Measurements

#### 4.3.1. Reaction 2 (Phosphoglucose Isomerase Reaction)

The determination of the enthalpy of reaction ${∆}_{R}H$ and kinetic parameters was done using ITC. The MicroCal PEAQ ITC from Malvern Panalytical (Malvern, UK) was used. For reaction 2 of the glycolysis, single measurements were performed with the F6P solution in the syringe and the PGI solution in the reaction cell. The reference cell was filled with water. The setup of the PEAQ-ITC was set to high feedback, reference power of 40 µW, stirrer speed of 750 rpm, and a titration speed of 0.5 µL s^−1^. To prevent the heat signal from being affected by premature diffusion of the substrate solution, two injections were performed, the first injection with 0.4 µL and a baseline of 10 min, which was not included in the evaluation, and the second injection as a main injection with 10 µL. This results in concentrations of 14.3 nmol kg^−1^ PGI and 4.8 mmol kg^−1^ F6P in the cell. The measurements were terminated when the heat signal returned to the baseline indicating the end of the reaction. Triplicate measurements were performed. It was impossible to perform the more common multiple injection measurements because the reaction was too fast and no plateau as required could be reached between the individual injections. The use of less enzyme to slow down the reaction was not an option either, as this would have made the instrumental thermal power change too small to obtain valuable kinetic data. In addition, multiple injection measurements only provide the initial rates of the reactions and do not take into account the near equilibrium that is the subject of this work. In order to calculate the influence of the heat of dilution, reference experiments were carried out and subtracted from the measurement curves. Two reference measurements were performed in which the enzyme or substrate solution were exchanged for buffers (Figure 2A) [53]. It was found that the titration of F6P to the buffer provides a large heat of dilution, whereas the titration of the buffer to PGI is negligible. The reference signal was then subtracted from the heat signal and the net signal of the reaction was received.

#### 4.3.2. Reaction 9 (Enolase Reaction)

For reaction 9 of the glycolysis, single injection measurements were performed, with an enolase solution in the titration syringe and a PEP solution in the sample cell. Since the heat measured during the reaction was very low, the highly concentrated PEP solution was added to the titration cell in order to allow the enzyme to convert a larger amount of PEP. Therefore, it was also impossible to do multiple injection measurements because a second injection of the enzyme does not change the equilibrium. The reference cell was filled with water. The setup of the PEAQ-ITC was set to high feedback, reference power of 41.9 µW, stirrer speed of 750 rpm, titration speed of 0.5 µL s^−1^, baseline recording of 15 min, and an injection volume of 39.2 µL. This means that the cell contains concentrations of 2 µmol kg^−1^ enolase and 74.8 mmol kg^−1^ PEP. The signal was recorded until it reached the baseline. The reference measurement was done with a buffer in the titration syringe and a PEP solution in the sample cell to measure the heat of dilution [53] (Appendix A). The reference signal was then subtracted from the signal of the reaction. Triplicate measurements were performed.

### 4.4. Determination of Reaction Enthalpy and Equilibrium Constant $K_{c}$

The initial range of the measurements were influenced by the thermal inertia of the instrument and the heat of dilution and was, therefore, not included in the evaluation. The reaction enthalpy $\Delta_{R}H$ was calculated (Equation (23)) from the observed heat production rate *P* (see also Figure 2B) [35]:(23)$${∆}_{R}H\;=\;\frac{\int_{0}^{\infty }P\left(t\right)\mathrm{dt}}{\left(c_{S}^{0}{-c}_{S}^{\mathrm{eq}}\right)*m\;}$$
where $c_{S}^{0}$ is the substrate concentration (in mol kg^−1^) after the injection and *m* is the mass of the reaction volume in the calorimetric vessel (in kg). $c_{S}^{eq}$ is the substrate concentration at the equilibrium (in mol kg^−1^) and was calculated from the apparent concentration-based equilibrium constant $K_{c}$. $K_{c}$ is defined in Equation (24):(24)$$K_{c}\;=\;\frac{c_{P}^{\mathrm{eq}}}{c_{S}^{\mathrm{eq}}}$$
(25)$$\mathrm{for}\;\mathrm{reaction}\;2:\;K_{c}=\;\frac{c_{F6P}^{\mathrm{eq}}}{c_{G6P}^{\mathrm{eq}}}$$
(26)$$\mathrm{for}\;\mathrm{reaction}\;9:\;K_{c}=\;\frac{c_{\mathrm{PEP}}^{\mathrm{eq}}\cdot c_{H_{2}O}^{\mathrm{eq}}}{c_{2\mathrm{PG}}^{\mathrm{eq}}}$$
with the equilibrium concentration of product $c_{P}^{eq}$ (in mol kg^−1^), the equilibrium concentration of F6P $c_{F6P}^{eq}$ (in mol kg^−1^), the equilibrium concentration of G6P $c_{G6P}^{eq}$ (in mol kg^−1^), the equilibrium concentration of PEP $c_{PEP}^{eq}$ (in mol kg^−1^), the equilibrium concentration of water $c_{H_{2}0}^{eq}$ (in mol kg^−1^), and the equilibrium concentration of 2PG $c_{2PG}^{eq}$ (in mol kg^−1^). In this work, $K_{c}$ values were not determined experimentally for two reasons. On the one hand, the reaction volume of approximately 250 µL is very small and would require a high dilution of the sample, which would lead to a large dilution error. On the other hand, it is necessary to keep the sample at a constant measuring temperature after the measurement to prevent a bias by the temperature dependency of the $K_{c}$ value. Therefore, $K_{c}$ was calculated from the thermodynamic equilibrium constant $K_{a}$ from previous works [40,42]. The ePC-SAFT parameters were fitted to the reaction-independent equilibrium data from large reaction volumes:(27)$$K_{a\;}{=\;K}_{c}\cdot {\;K}_{\gamma }$$

In contrast to $K_{c}$ and $K_{\gamma }$, which for example depend on the substrate and product concentrations, $K_{a}$ is a constant that only depends on temperature, pressure, and pH. To determine $K_{c}$ at different reaction conditions, ePC-SAFT was applied. A brief description of ePC-SAFT can be found in the Appendix A. Starting with the initial substrate concentration, the progression of the PGI reaction was simulated by stepwise decreasing the substrate and increasing the product concentration. For each step, $K_{\gamma }$ was predicted with ePC-SAFT from the activity coefficients of the substrate and the product. All substances present in the reaction solution, except the enzyme, were considered for the determination of the activity coefficients. The $K_{a}$ values of each step, calculated from the respective $K_{c}$ and $K_{\gamma }$ were compared to the known $K_{a}$ value and the iteration was continued until both were the same. Rational activity coefficients $\gamma_{i}^{*}$ were used for the reacting agents because they are highly diluted in water. Thus, the activity-coefficient ratio $K_{\gamma }$ is calculated according to Equation (28):(28)$$K_{\gamma }\;=\;\frac{\gamma_{P}^{*,m,\mathrm{eq}}}{\gamma_{S}^{*,m,\mathrm{eq}}}$$

In this work, the hypothetical ideal solution is defined as a solution of 1 mol kg^−1^ of the substance diluted in water and an activity coefficient equal to that of the substance infinitely diluted in water (i.e., $\gamma_{i}^{*}=1$).

### 4.5. Kinetic Investigations

In our work, the opposite direction of the glycolysis was investigated for practical reasons. For reaction 2, the backward reaction from F6P to G6P was measured, as it was evident from literature data that the reaction equilibrium is on the side of G6P [42]. For reaction 9, the backward reaction from PEP to 2PG was investigated because 2PG was only available as barium salt, which could influence the kinetic parameters obtained due to the high concentrations used. However, backward reactions are frequently investigated in the literature [21,24,43] since the direction is irrelevant for the determination of reaction rates (Equation (29)).

For testing kinetic models, reaction rates as a function of the substrate concentration are required. The concentration of the substrate at each single measuring point was calculated with Equation (29) using the concept of Todd and Gomez [35]:(29)cS(t) = cS0−∫0tP(t)·dt(∆HR ·m)
where *P*(*t*) represents the heat production rate (in W). The corresponding rate $r_{S}$ correlates to the heat production rate according to Equation (30):(30)rS(t) = P (t)∆HR *m

The concentration of the substrate and the rate of the substrate consumption were plotted against each other. The first seconds before the heat flow reaches its maximum were not included in the data evaluation because initially both the inertia of the ITC and the heat of dilution influence the signal.

### 4.6. Statistics

For the data evaluation and statistics we used OriginPro, Version 2019 OriginLab Corporation, Northampton, MA, USA. A statistical evaluation was carried out for all the measurements. Measurements at every temperature were performed as triplicates. For each parameter the mean and the standard deviation were reported. Noor′s fit parameters provided an R^2^ value above 0.98 (except for one experiment). For the flux-force model, both an R^2^ and a *p*-value could be determined. The *p*-value is given as zero by Origin and at a 0.05 level slope is significantly different from 0. In every flux-force measurement the R^2^ value was greater than 0.993. The exact values can be found in Appendix A.

## 5. Conclusions

Systems biology approaches aim for the simplest possible kinetic approaches with as few adjustable parameters as possible, while maintaining a good representation of the kinetics at different cytosolic conditions. The conventional irreversible Michaelis-Menten model has only two parameters but does not take the backward reaction into account. The reversible Michaelis-Menten equation and models derived from it capture the equilibrium correctly, but often have too many parameters to describe the influence of the cytosolic conditions in a simple manner. The Noor model and the one-parameter flux-force relations seem to be perfectly suited for this purpose and to describe the time course of at least reactions 2 and 9 of glycolysis. An advantage of flux-force approaches is, that they combine the thermodynamics and kinetics of metabolic processes in a single equation in order to quantitatively describe the approximation to the equilibrium perfectly. Both models tested, Noor and flux-force, show similar dependencies of the reaction rate on the temperature. The flux-force model could also be verified using an Arrhenius plot. Further work will be done to verify these statements under cytosolic conditions. In other glycolytic reactions or initial conditions at an equilibrium distance with very large thermodynamic driving forces, deviations from the observed linear flux-force relationship may occur. For reaction cascades with complex interactions even oscillations are described. This will be further investigated in future work.

The ITC is well suited for a fast and easy quantitative analysis. Especially in the non-linear range with the possibility of oscillations, the heat production rate as a real time signal could be very useful. Time-consuming and failure-prone sampling is no longer necessary. However, the reaction time should be chosen long enough to assure that the kinetics of the reaction rather than the thermal inertia of the instrument determine the signal. This can be achieved by wisely selecting a suitable enzyme concentration, for example.

## Figures and Tables

**Figure 1 ijms-21-08341-f001:**
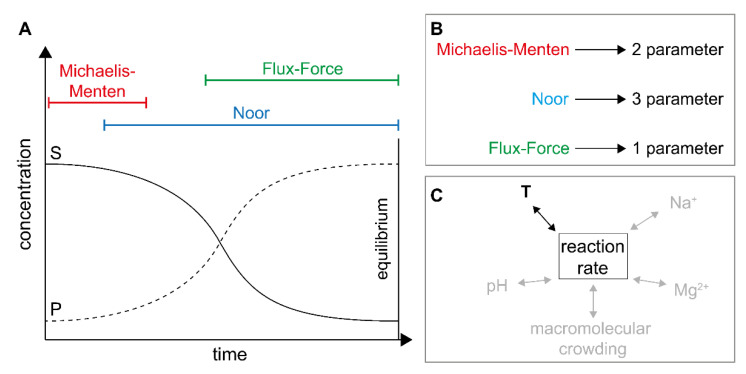
Scheme of the investigations within the present work. (**A**) Shows exemplarily the validity range of the various kinetic models used to describe the reaction progress, (**B**) shows the number of adjustable parameters required by the models, and (**C**) shows the most important cytosolic conditions, highlighting the temperature as the one used to test the models.

**Figure 2 ijms-21-08341-f002:**
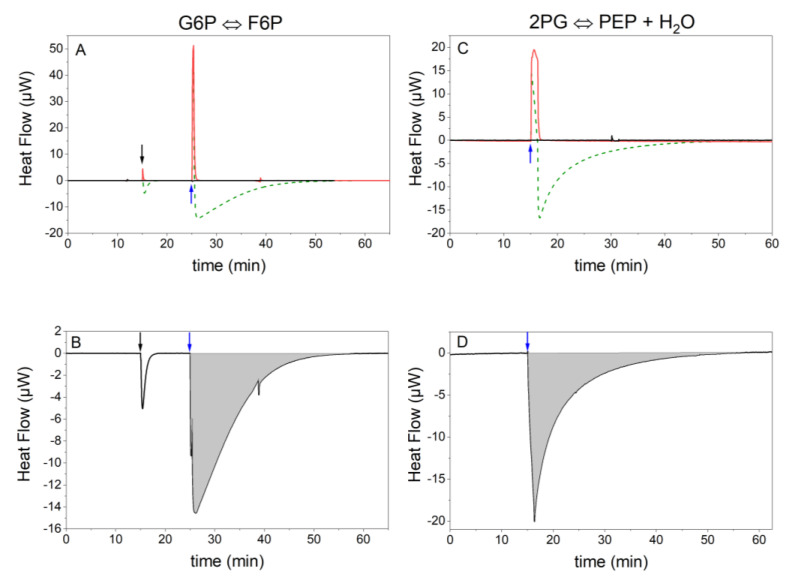
Heat flow diagrams from the isothermal titration calorimetry (ITC) measurement of reactions 2 and 9 at 310.15 K. The arrows mark the time of the two injections (black = 1st injection, blue = 2nd injection). At the end of the reaction, when the equilibrium is reached, the signal returns to the baseline. The concentrations were 14.3 nmol kg^−1^ PGI and 4.8 mmol kg^−1^ F6P for reaction 2 and 2 µmol kg^−1^ enolase and 74.8 mmol kg^−1^ PEP for reaction 9. (**A**,**C**) Show the heat flow curves of the total reaction signal (green), the reference measurement without the enzyme (red), and the reference measurement without the substrate (black). The heat of dilution causes the positive peak of the red and green curve. (**B**,**D**) Display the net reaction (subtraction of the reference signals from the total reaction signal). The integrated heat of the net signal (Q) is shown in gray.

**Figure 3 ijms-21-08341-f003:**
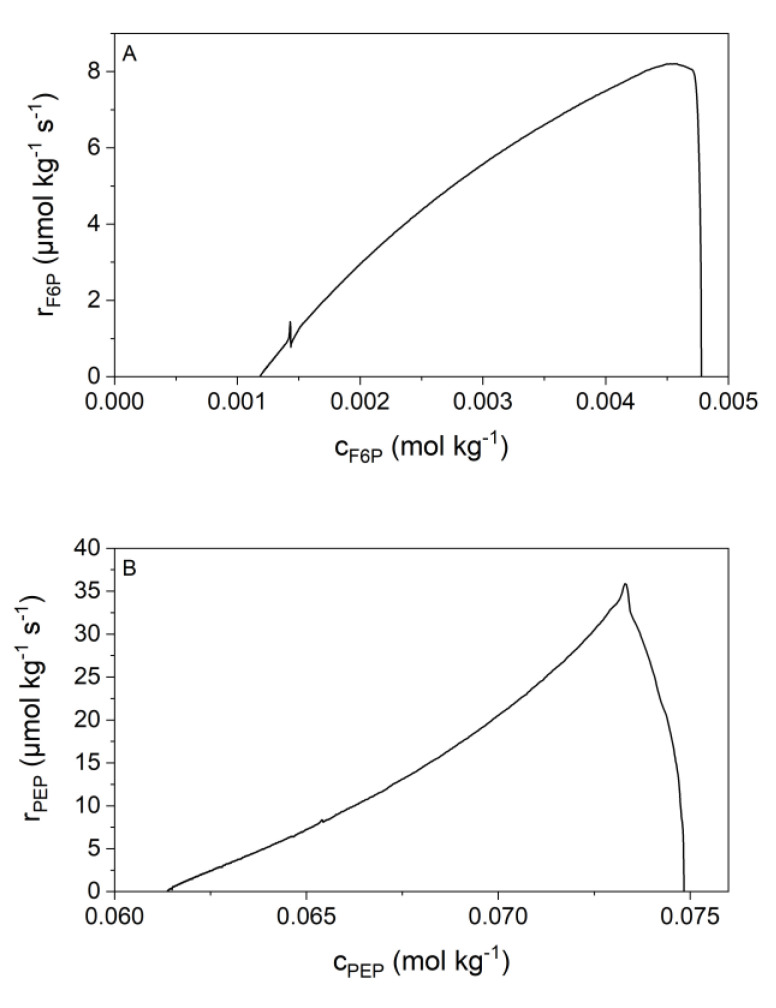
Plot of reaction rate against the substrate concentration for both reactions at 310.15 K. Both panels show a single example reaction. (**A**) Displays reaction 2 with a concave curvature but no plateau at high substrate concentrations (14.3 nmol kg^−1^ PGI and 4.8 mmol kg^−1^ F6P). (**B**) Reaction 9 with a convex curvature, also not showing a plateau at high substrate concentrations (2 µmol kg^−1^ enolase and 74.8 mmol kg^−1^ PEP).

**Figure 4 ijms-21-08341-f004:**
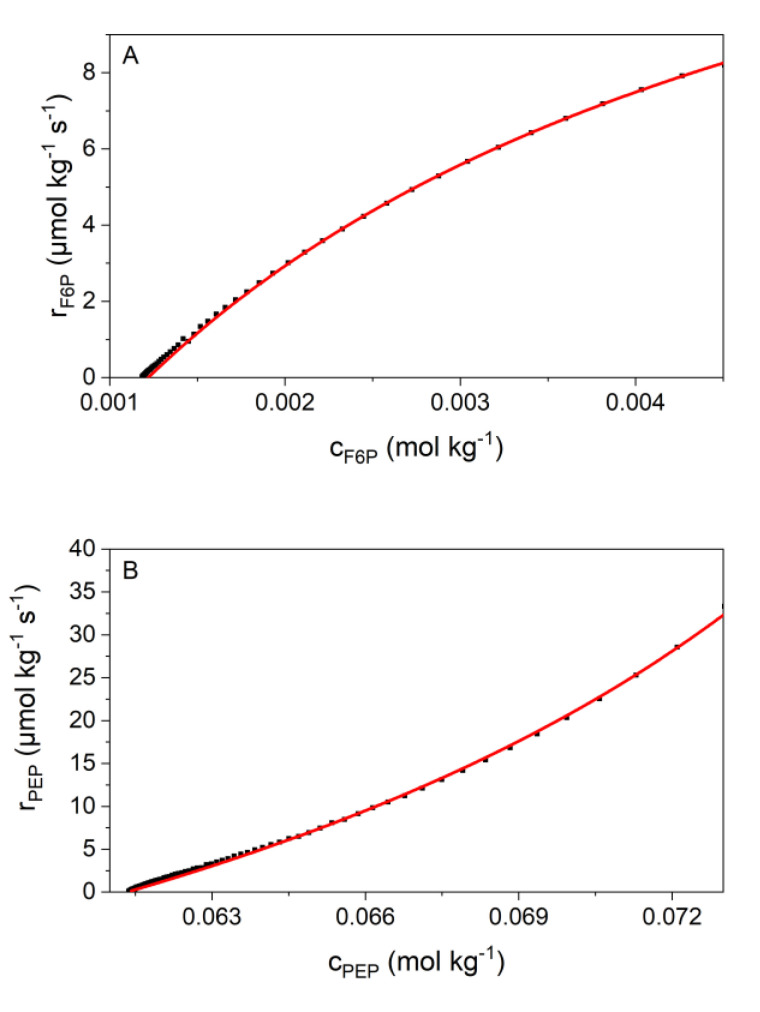
Fitting of the kinetic data (scatter) with the Noor model (red solid line). Both panels show a single example reaction. (**A**) Shows the data of reaction 2 with concentrations of 14.3 nmol kg^−1^ PGI and 4.8 mmol kg^−1^ F6P (R^2^ = 0.99972). (**B**) Shows the data of reaction 9 with concentrations of 2 µmol kg^−1^ enolase and 74.8 mmol kg^−1^ PEP (R^2^ = 0.99903). Both were measured at 310.15 K. The adjustable parameters are $r_{max}$ and $K_{F6P}$ in A and Λ and $K_{2PG}$ in B.

**Figure 5 ijms-21-08341-f005:**
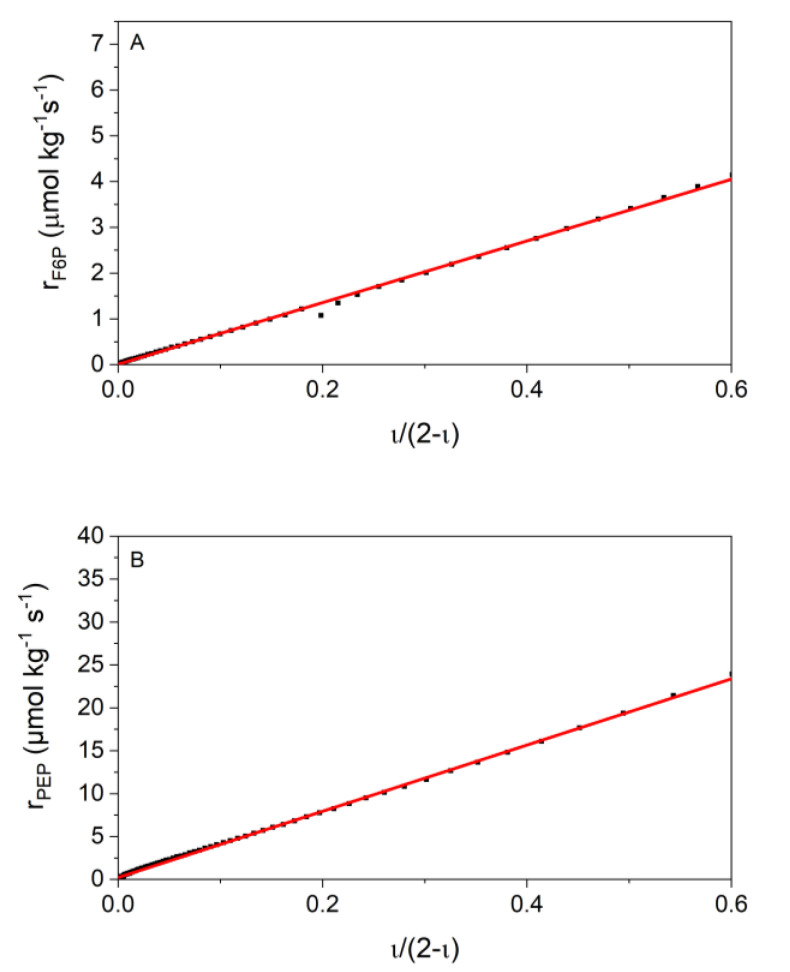
Verification of the suitability of the flux-force model to describe the reaction kinetics at 310.15 K. Both panels show a single example reaction. (**A**) Shows the result of one chosen condition of reaction 2 (14.3 nmol kg^−1^ PGI and 4.8 mmol kg^−1^ F6P) with R^2^ = 0.99921 and (**B**) of reaction 9 with concentrations of 2 µmol kg^−1^ enolase and 74.8 mmol kg^−1^ PEP (R^2^ = 0.99892).

**Figure 6 ijms-21-08341-f006:**
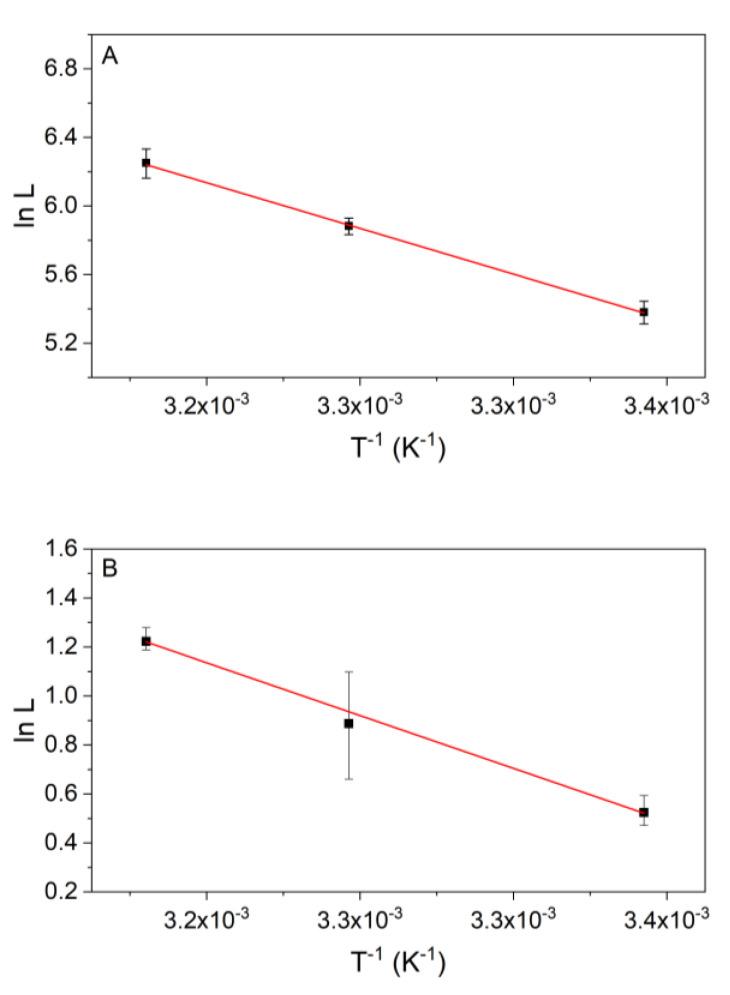
The Arrhenius plot of the kinetic parameter *L* (flux-force model) for both reactions. (**A**) Shows results for reaction 2 (14.3 nmol kg^−1^ PGI and 4.8 mmol kg^−1^ F6P) with a *p*-value of the slope of 0.01389 (slope is significantly different from 0) and R^2^ = 0.99952 and (**B**) for reaction 9 with concentrations of 2 µmol kg^−1^ enolase and 74.8 mmol kg^−1^ PEP (*p*-value = 0.01922 (slope is significantly different from 0), R^2^ = 0.99909). The error bars correspond to the standard deviations of the triple determinations. The activation energy $E_{a}$ was calculated from the slope of the fit. For reaction 2, it is $E_{a}\;$ = 55.4 ± 1.2 kJ mol^−1^ and for reaction 9, it is $E_{a}\;$ = 44.8 ± 1.4 kJ mol^−1^.

**Table 1 ijms-21-08341-t001:** Symbols.

Symbol	Property	Unit
*A*	pre-exponential factor	s^−1^
$a_{i}$	activity of component i	-
$c_{i}$	concentration of component i	mol kg^−1^
$c_{i}^{0}$	concentration of component i at time 0	mol kg^−1^
$c_{i}^{eq}$	equilibrium concentration of component i	mol kg^−1^
$c_{P}$	product concentration	mol kg^−1^
$c_{S}$	substrate concentration	mol kg^−1^
$E_{a}$	activation energy	J mol^−1^
$\Delta_{R}G$	Gibbs energy of biochemical reaction	J mol^−1^
$\Delta_{R}G^{o}$	standard Gibbs energy of biochemical reaction	J mol^−1^
$\Delta_{R}H$	enthalpy of biochemical reaction	J mol^−1^
*J*	flux	mol L^−1^ s^−1^
$K_{a}$	thermodynamic equilibrium constant of biochemical reaction	-
$K_{c}$	apparent equilibrium-molality ratio of biochemical reaction	−/mol kg^−1^
$k_{cat}$	kinetic constant of reaction	s^−1^
$K_{S/P}$	Michaelis constant for substrate/product	mol kg^−1^
$K_{\gamma }$	activity-coefficient ratio of biochemical reaction	−/mol kg^−1^
*L*	phenomenological coefficient/kinetic parameter	s^−^^1^
*m*	mass	kg
*P*	heat production rate	W
*Q*	heat	J
R	universal gas constant (8.314 J mol^−1^ K^−1^)	J mol^−1^ K^−1^
r	reaction rate	mol L^−1^ s^−1^
$r_{max}$	maximum reaction rate	mol L^−1^ s^−1^
T	temperature	K
*α*	correlation factor	mol kg^−1^ s^−1^
Λ	kinetic parameter $\Lambda \;=\;\;\frac{r_{max}}{K_{S}}$	s^−1^
ι	thermodynamic driving force	-
$\gamma_{i}^{*,m}$	rational activity coefficient of component i on molality base	-

**Table 2 ijms-21-08341-t002:** Reaction enthalpy values of reaction 2 for different buffer systems.

Buffer	$\Delta_{R}H\;(\mathrm{kJ}\;{\mathrm{mol}}^{-1})$
HEPES	9.7 ± 0.3
Potassium phosphate	9.6 ± 0.2
MOPS	11.1 ± 0.5

**Table 3 ijms-21-08341-t003:** Measured thermodynamic reaction properties.

	Reaction 2			Reaction 9 [40]		
Temperature (K)	Q (mJ)	$K_{c}\;(\text{–})$	$\Delta_{R}H\;\;(\mathrm{kJ}\;{\mathrm{mol}}^{-1})$	Q (mJ)	$K_{c}\;(\mathrm{mol}\text{·}\mathrm{kg}{-}^{1})$	$\Delta_{R}H\;\;(\mathrm{kJ}\;{\mathrm{mol}}^{-1})$
298.15	7.63 ± 0.14	0.285	10.3 ± 0.2	8.15 ± 0.72	239.4	2.4 ± 0.2
305.15	7.63 ± 0.07	0.318	10.6 ± 0.1	7.85 ± 0.17	245.9	2.4 ± 0.1
310.15	7.86 ± 0.33	0.343	11.1 ± 0.5	7.64 ± 0.22	251.3	2.4 ± 0.1

The error bars result from the standard deviation of the triple determinations. Results for reaction 9 are adapted from [40].

**Table 4 ijms-21-08341-t004:** Obtained Noor parameters for reactions 2 and 9 at different temperatures. The error bars correspond to the standard deviations of the triple determinations.

Temperature (K)	Reaction 2		Reaction 9	
	$r_{max}$ (µmol kg^−1^s^−1^)	$K_{F6P}$ (mmol kg^−1^)	Λ (ms^−1^)	$K_{2PG}$ (mmol kg^−1^)
298.15	7.03 ± 0.58	4.35 ± 0.64	0.34 ± 0.01	10.2 ± 0.92
305.15	10.30 ± 0.88	3.41 ± 0.39	0.44 ± 0.13	13.8 ± 3.35
310.15	13.21 ± 1.20	3.26 ± 0.09	0.57 ± 0.03	16.5 ± 0.48

**Table 5 ijms-21-08341-t005:** Results from the flux-force model for reactions 2 and 9 and the two glycolytic reactions at different temperatures. The flux-force model provides values for *L* from the slope of the plot (Figure 5). The error bars correspond to the standard deviations of the triple determinations.

Temperature (K)	Reaction 2	Reaction 9
	*L* Value (s^−1^)	*L* Value (s^−1^)
298.15	217.18 ± 14.27	10.13 ± 0.65
305.15	358.80 ± 17.29	14.57 ± 3.20
310.15	518.44 ± 43.84	20.37 ± 1.03

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
