# Peer review of "Thermodynamics and Kinetics of Glycolytic Reactions. Part I: Kinetic Modeling Based on Irreversible Thermodynamics and Validation by Calorimetry"

_ijms, 2020, doi:10.3390/ijms21218341_

Round 1

Reviewer 1 Report

Reviewer’s Report :

Manuscript ID: ijms-930366 

Title: Thermodynamics and kinetics of glycolytic reactions.

Part I: Kinetic modeling based on irreversible thermodynamics and validation by calorimetry

Journal: International Journal of Molecular Sciences

This manuscript reports thermodynamic and kinetic investigation of glycolytic reaction steps 2 (the phosphoglucose isomerase reaction from glucose6-phosphate to fructose6-phosphate) and 9 (the enolase-catalyzed reaction from 2-phosphoglycerate to phosphoenolpyruvate and water) under cytosolic conditions. While the topic of research is interesting in system-biology network analysis, only the linear thermodynamic  flux-force relations have been included in this manuscript. Glycolysis involvs linear as well as nonlinear processes between flux and thermodynamic force including the glycolytic oscillations (viz. Hess et al. Z. Naturforsch. 40C(1985), 588 ; Sel'kov, Eur. J. Biochem. 4 (1968), 79). The novelty of this work has been greatly diminished because of the absence of more interesting nonlinear processes of glycolytic reactions.

I have faced the following problems while reading the manuscript.

  1. i)  Confusions from double numbering : reaction 2 (also 4) and reaction 9 (also 5).
  2. ii) l-part of equation (9) is incorrect after substitution of Cp = Cs°-Cs in equation (7b). The correct value of l-part of equation (9) seems to be :

1 - [Cs(eq).(Cs° - Cs) / (Cs° - Cs(eq).Cs]

iii) Equation (13) should be written correctly as

r+ + r- = L.C(E)                                                                                                     (13)

  1. iv) Because of the presence of nonlinearity, Arrhenius plots of ln(rate constant) vs. 1/T should be different from a straight line and the calculation of activation energy from such plots does not arise.

 I am unable to recommend publication of this manuscript in Int.J. Mol.Sci.in the present form.

Author Response

please see the attached reply

Reviewer 2 Report

This manuscript presents a potentially useful approach that may help develop better kinetic models of metabolism. The topic is of interest and aproppiate for this journal. However, before further consideration for publication, the authors must address the following moderate and minor concerns:

Specific comments:

54 – grammar: add an ‘and’ after substrate and remove comma

99 – Provide the full name of the acronym DADPH/LDH.

144-145 – Why did you evaluate the opposite direction of reactions 2 and 9?

198 – How is  (the pre-exponential factor constant) calculated?

207 – Provide the full name for the acronym SI.  

214 – Increase the font size of the ‘A’ and ‘B’ lettering on Figure 2.

214 – There are multiple green and red lines on Figure 2A. What is the significance of the two small red peaks and the small green peak on both ends?

223 – You conclude that the ionization of the buffer does not play a significant role in the reaction enthalpy because your calculated values for reaction 2 were 9.6, 9.7 and 11.1; however, I think it’s important to mention that it does play some role and that ionic strength cannot be ignored when calculating these values.

229 – Provide the full name for the acronym ePC-SAFT. You should also mention that a full explanation of the ePC-SAFT methods are provided in the supplemental information.

256-258 – There are references to equations 15a and 15b but only a single equation is provided for equation 15.

535-538 – Is there evidence that suggests that most metabolic processes probably take place close to equilibrium? This statement can potentially be misleading because there are plenty of examples of reactions that take place far from equilibrium.

Additional general issues:

  • Some references have an error associated with them and that needs to be corrected (i.e. Error! Reference source not found).
  • Increase the font size of the equations and center them.
  • Fix general typos and grammar errors present throughout the paper.
  • In the methods section, keep your equation formatting consistent. For examples, with respect to the enthalpy of the reaction, the formatting of this variable in the methods section differs from the formatting seen throughout the paper (e.g. lines 434, 470)
  • In Table S2, you mention that the ePC-SAFT parameters you use for 2-PG are based on 3-PG. This should be mentioned in the discussion considering you use ePC-SAFT to predict  values.

Author Response

Please see the attached reply.

Reviewer 3 Report

This manuscript discusses the use of irreversible thermodynamics to describe reversible enzyme-catalyzed reactions. The authors employed the Noor model and the flux-force model to describe the near-equilibrium reaction kinetics of reactions 2 and 9 of glycolysis that irreversible Michaelis-Menten kinetics cannot explain. The benefit of the flux-force approach over the Noor approach is that it requires only one parameter and works reasonably well for near-equilibrium reactions. The authors claim that the flux-force approach is better for investigating the influence of cytosolic conditions. The researchers employed ITC to measure reaction enthalpies and derive reaction rates. The temperature dependence of the kinetic parameter satisfied the Arrhenius relation and resulted in plausible activation energies. While the authors provide useful simplification for describing reaction kinetics, it is doubtful that the ideas presented in this manuscript are novel or widely applicable. The paper may be accepted after adding more examples and addressing the comments below.

Major comments:

  1. How does the flux-force approach take into consideration various small molecule regulation or post-translational modification?
  2. Please describe how reversible Michaelis-Menten kinetics and the Noor model are different.
  3. A major weakness of this paper is its lack of unique contribution. Their approach involved applying two preexisting theories.
  4. Please specify dG range that the authors consider as “near-equilibrium.” There are other and perhaps closer-to-equilibrium glycolytic reactions: what is the rationale behind choosing PGI and ENO? This paper would be stronger if it encompassed all “near-equilibrium” glycolytic steps. Currently, there are not enough data or examples of other glycolytic reactions to conclude that the flux-force approach can be used broadly.

Minor comments:

  1. The introduction should be more concise.
  2. Line 164: Please explain why the single-saturation assumption is suitable
  3. Error in eqn. 13: Remove “r=” because variable ‘r’ in eqn. 12 and ‘r’ in eqn. 13 seem to be different.
  4. Is the ‘L’ in eqn. 6 different from the ‘L’ in eqn. 13? If so, please use different variable letters.
  5. Line 188: “comparing the influence of cytosolic conditions on the kinetics” is vague. Why is the parameter L suitable for that? What methods/experiments could be done to test other cytosolic conditions and their effect on the model?
  6. 2: How do the green line in panel A differ from the black line in panel B? In panel A, why does green line go above 0 at ~25 mins in panel A? Does this show one experiment or the mean of replicates? It would be helpful if the legends explained what is happening at the different times in the ~1 hr ITC measurement period.
  7. Table 1 shows statistically significant differences between MOPS and the other two buffers although the authors concluded that the enthalpy values were all similar.
  8. Line 256: Where are equations 15a and 15b?
  9. 3: please explain why the rates drop to 0 at ~4.5 mmol/kg F6P and at ~73 mmol/kg PEP.
  10. Please show the goodness of fit from the Noor model, the flux-force model, and Michaelis-Menten equation.
  11. Fix all “Error! Reference source not found.”

Author Response

Please see the attached reply.

Reviewer 4 Report

This manuscript describes the kinetic modeling of the reversible reactions in the glycolysis such as Pgi and Eno based on the irreversible thermodynamics and validation by calorimetry.

  1. English should be checked throughout the manuscript. What does ‘Error Reference source not found’ mean in pages 3, 4, 7, 8, 9, 10, 13, 14, and 15 ?
  2. Statistical analysis method should be given in Materials and method section, and any comparison must be made based on the statistical analysis with p-value. Otherwise, no statistically meaningful discussion can be made.
  3. It is not clear why Pgi and Eno were chosen, where the next pathways are the irreversible rate-limiting pathways such as Pfk and Pyk, respectively ?
  4. The equilibrium state is reached when the glycolytic and gluconeogenic pathway reactions coexist ?
  5. I wonder why r(F6P) drops at higher C(F6P), and r(PEP) drops at higher C(PEP) in Fig.3 ? The r may be given as r(Pgi) rev and r(Eno) rev ?
  6. The r(F6P) and r(PEP) may be also functions of C(G6P) and C(2PG) in Fig.3 ?
  7. What is the physiological meaning of the concave cube of r(PEP) with respect to C(PEP) instead of convex ?
  8. [mol kg-1] in Figs.3 and 4 may be changed to [mol g-1] ?
  9. Error bars should be given for the experimental data as shown in Figs. 3 and 4。
  10. The effects of the ate limiting steps such as Pfk and Pyk on the metabolism has been extensively investigated from the thermodynamic driving force point of view with ΔG. Here, instead, attention is focused on the reversible pathway reactions. What is the implication or impact of the present approach in understanding the physiology of the glycolysis ?

Author Response

Please see the attached reply.

Round 2

Reviewer 1 Report

Reviewer’s Report :

Manuscript ID: ijms-930366 –V2

Title: Thermodynamics and kinetics of glycolytic reactions.

Part I: Kinetic modeling based on irreversible thermodynamics and validation by calorimetry

Journal: International Journal of Molecular Sciences

This manuscript reports thermodynamic and kinetic investigation of glycolytic reaction steps 2 (the phosphoglucose isomerase reaction from glucose6-phosphate to fructose6-phosphate) and 9 (the enolase-catalyzed reaction from 2-phosphoglycerate to phosphoenolpyruvate and water) under cytosolic conditions.

*Theauthors are correct that c(E) in my comments about eq.(13) in my first review is to be replaced by cE (This was a typo).

*Show neatly the deduction of  eq.(11) from basic thermodynamic relations before you use it and eq.(6) to deduce eq.(12). I see a mathematical inaccuracy in these calculations.

*The systems investigated in the manuscript are essentially closed systems in thermodynamic sense (contrary to open systems as, for example, in Ross et al. Prog. Theor. Phys. 1981, 66, 385). In such closed systems nonlinearity can't go beyond an upper value, which essentially prevents bistability/oscillation  to appear. In closed systems, nonlinearity is maximum at the start of the experiment, which presumably decreases as the reaction progresses towards thermodynamic equilibrium. Very close to equilibrium, phenomenological coefficients become symmetric to establish a linear relation between flux and thermodynamic force. In that region, the authors seem to have undertaken experiments, and, therefore, their results including temperature effects experiments remain valid, which I may have wrongly interpreted  in my first review. But the authors must include in the text why nonlinearity is much less in their type of closed systems (in thermodynamic sense), so that ,their temperature effects experiments remain valid. This may vastly improve the quality of presentation in such a closed system  of  glycolytic experiment.

*line 310/397/398:  Eq.(20) should be replaced by eq.(15) ; Arrhenius should be replaced by Arrhenius equation.

*line 207/433 : Table S1 and fig.S1 should be included in the text or as an Appendix in the same file. This is a must to make the manuscript readable.

*line 561 : The statement ‘Many of the metabolic processes probably take place close to equilibrium’ included in Conclusion Section is incorrect, because metabolic processes in living systems are continuously fed open systems (in thermodynamic sense) with high nonlinearity.

I am unable to recommend the second version of this manuscript in the present form for publication in the Int.J.Mol. Sci.

Author Response

Many thanks for your helpful advixces. Please see our response in the attached file

Reviewer 3 Report

Remaining concerns:

  1. The response to reviewers and the clarification written in the point-by-point response should be incorporated into the manuscript. While the responses are satisfactory, the manuscript has not improved by much compared to the previous version.
  2. The relationship between r and L from equations 6 and 14 is still confusing. Can the authors define the “L” for both the flux-force model and the Noor model in the manuscript?
  3. On a related note, Eqn. 13 seems still wrong. In line 183, it is stated that the sum of r+ and r- and the amount of enzyme cE are proportional, yet Eqn. 13 shows the difference between r+ and r-.
  4. Line 183 contains unnecessary “- “; please remove this.
  5. Consistently use either “back reactions” or “backward reactions” throughout the manuscript.
  6. Please check for grammar, syntax, and sentence structure.

We recommend publication of the manuscript after the remaining concerns are addressed.

Author Response

Many thanks for the helpful advices. Please find attached a file with a point-by-point response.

Round 3

Reviewer 1 Report

Reviewer’s Report :

Manuscript ID: ijms-930366 –V3

Title: Thermodynamics and kinetics of glycolytic reactions.

Part I: Kinetic modeling based on irreversible thermodynamics and validation by calorimetry

Journal: International Journal of Molecular Sciences

This manuscript reports thermodynamic and kinetic investigation of glycolytic reaction steps 2 (the phosphoglucose isomerase reaction from glucose6-phosphate to fructose6-phosphate) and 9 (the enolase-catalyzed reaction from 2-phosphoglycerate to phosphoenolpyruvate and water) under cytosolic conditions.

*I have discovered a mathematical inaccuracy in the derivation of eq.13 (version3) ; the final result  l/(2-l) is correct, but the intermediate step

is incorrect, which makes the final result negative of what is written above (very very important !!).

*Table S1 and fig.S1 should be included in the text file, which will help reading of the manuscript without interruptions. It's okay if other supplementary materials are put in a different file for specially interested readers.

*Although the manuscript is lacking novelty, the experimental data is required to be published. The presentations must be in brief

because all these are already published materials from other groups.

*The manuscript title/abstract/conclusion must reflect that this work is a contribution, near thermodynamic equilibrium only, in the backdrop that the whole nonlinear region of the glycolytic pathway involving asymmetric Onsager coefficients has  been ignored.

*Publication of the manuscript is recommended only after the inclusion of the above changes I have listed.

Round 4

Reviewer 1 Report

Reviewer’s Report :

Manuscript ID: ijms-930366 –V4

Title: Thermodynamics and kinetics of glycolytic reactions.

Part I: Kinetic modeling based on irreversible thermodynamics and validation by calorimetry

Journal: International Journal of Molecular Sciences

This manuscript reports thermodynamic and kinetic investigation of glycolytic reaction steps 2 (the phosphoglucose isomerase reaction from glucose6-phosphate to fructose6-phosphate) and 9 (the enolase-catalyzed reaction from 2-phosphoglycerate to phosphoenolpyruvate and water) under cytosolic conditions.

*It is suggested that Fig.S1 be included in the main manuscript file to support reading of the manuscript without interruption, while TableS1 may be left included within supplementary materials in a different file.

*The novelty of this work on glycolytic pathway and its limitation for possible extensions to future nonlinear domain investigations, has not been discussed clearly in the conclusion.The authors may wish to add a few more lines regarding this.

*The manuscript is recommended for publication, if other reviewers do not object.
